# Endocannabinoid 2-Arachidonoylglycerol Levels in the Anterior Cingulate Cortex, Caudate Putamen, Nucleus Accumbens, and Piriform Cortex Were Upregulated by Chronic Restraint Stress

**DOI:** 10.3390/cells12030393

**Published:** 2023-01-21

**Authors:** Qing Zhai, Ariful Islam, Bin Chen, Hengsen Zhang, Do Huu Chi, Md. Al Mamun, Yutaka Takahashi, Noriko Sato, Hidenori Yamasue, Yoshiki Nakajima, Yu Nagashima, Fumito Sano, Tomohito Sato, Tomoaki Kahyo, Mitsutoshi Setou

**Affiliations:** 1Department of Cellular and Molecular Anatomy, Hamamatsu University School of Medicine, Hamamatsu 431-3192, Shizuoka, Japan; 2Preppers Co., Ltd., Hamamatsu University School of Medicine, Hamamatsu 431-3192, Shizuoka, Japan; 3International Mass Imaging Center, Hamamatsu University School of Medicine, Hamamatsu 431-3192, Shizuoka, Japan; 4Department of Anesthesiology, Seirei Mikatahara General Hospital, Hamamatsu 433-8558, Shizuoka, Japan; 5Department of Psychiatry, Hamamatsu University School of Medicine, Hamamatsu 431-3192, Shizuoka, Japan; 6Department of Anesthesiology and Intensive Care, Hamamatsu University School of Medicine, Hamamatsu 431-3192, Shizuoka, Japan; 7Institute for Medical Photonics Research, Preeminent Medical Photonics Education and Research Center, Hamamatsu University School of Medicine, Hamamatsu 431-3192, Shizuoka, Japan; 8Department of Systems Molecular Anatomy, Institute for Medical Photonics Research, Preeminent Medical Photonics Education & Research Center, Hamamatsu 431-3192, Shizuoka, Japan

**Keywords:** endocannabinoid 2-arachidonoylglycerol, brain, chronic restraint stress, DESI-MSI

## Abstract

Endocannabinoid 2-arachidonoylglycerol (2-AG) has been implicated in habituation to stress, and its augmentation reduces stress-induced anxiety-like behavior. Chronic restraint stress (CRS) changes the 2-AG levels in some gross brain areas, such as the forebrain. However, the detailed spatial distribution of 2-AG and its changes by CRS in stress processing-related anatomical structures such as the anterior cingulate cortex (ACC), caudate putamen (CP), nucleus accumbens (NAc), and piriform cortex (PIR) are still unclear. In this study, mice were restrained for 30 min in a 50 mL-centrifuge tube for eight consecutive days, followed by imaging of the coronal brain sections of control and stressed mice using desorption electrospray ionization mass spectrometry imaging (DESI-MSI). The results showed that from the forebrain to the cerebellum, 2-AG levels were highest in the hypothalamus and lowest in the hippocampal region. 2-AG levels were significantly (*p* < 0.05) upregulated and 2-AG precursors levels were significantly (*p* < 0.05) downregulated in the ACC, CP, NAc, and PIR of stressed mice compared with control mice. This study provided direct evidence of 2-AG expression and changes, suggesting that 2-AG levels are increased in the ACC CP, NAc, and PIR when individuals are under chronic stress.

## 1. Introduction

Endocannabinoid 2-arachidonoylglycerol (2-AG) is one of the main endogenous cannabinoid ligands in the central nervous system [1]. It is a neutral lipid that has the intrinsic tendency to associate with membranes, synthesize “on demand” [2], release from postsynaptic neurons, and travel backward across synapses, activating the CB1 receptor on presynaptic axons and suppressing neurotransmitter release, functioning as a retrograde synaptic messenger [3]. 2-AG could be biosynthesized from 1-oleoyl-2-arachidonoyl-sn-glycerol (OAG) [4] and 1-stearoyl-2-arachidonoylglycerol (SAG) [5] by diacylglycerol lipases (DAGLs) in the brain tissues. DAGLs are considered the most important enzymes for 2-AG biosynthesis, and monoacylglycerol lipase (MGL) is a key enzyme for 2-AG hydrolysis [2]. 2-AG displays anti-inflammatory and neuroprotective properties [6]. It also participates in responses to stress stimuli and is becoming a potential new therapeutic target in treating major depressive disorder (MDD) [7]. It might act to mitigate depressive symptoms [8], reduce stress-induced anxiety-like behavior [9], and participate in habituation to repeated stress [10]. One study has suggested an association between stress-related disorders and an increased risk of neurodegenerative diseases, such as Alzheimer’s disease [11]. All the information suggests that 2-AG is an essential lipid messenger in the brain. Revealing the spatial distribution of 2-AG in brain tissue will be beneficial for the study of related diseases, such as stress-related disorders and neurodegenerative diseases.

Stress is a risk factor for the development and exacerbation of many diseases, such as neuropsychiatric disorders [12]. Chronic restraint stress (CRS) is a preclinical chronic stress exposure model for investigating stress-relevant disorders [13]. It has been reported that 2-AG levels were significantly increased in the forebrain after 5 days of CRS exposure [14] and in the medial prefrontal cortex (mPFC) after 10 days of CRS exposure [10]. These results were measured by liquid chromatography-mass spectrometry (LC-MS). However, LC-MS could not provide the spatial information of 2-AG, and it is difficult to apply to small brain anatomical structures. The mPFC is a part of the forebrain, containing the anterior cingulate cortex (ACC), prelimbic cortex, and infralimbic cortex. ACC is associated with stress-related illnesses, such as MDD [15], anxiety, and mood disorders [16]. Extensive preclinical evidence has also established the mediating role of the caudate putamen (CP) [17], nucleus accumbens (NAc) [18], and piriform cortex (PIR) [19] in stress processing or stress-related disorders. However, the detailed spatial distribution of and changes in 2-AG after CRS exposure in these regions are still unclear.

Mass spectrometry imaging (MSI) can provide complementary information for molecular analysis and is applied by the scientific community to study proteins, peptides, small molecules, and metabolites [20]. Desorption electrospray ionization MSI (DESI-MSI) is a technique that allows the spatial intensity distribution to be recorded directly from histological sections without prior chemical treatment under ambient conditions. It can simultaneously visualize target molecules, such as metabolites [21] and lipids [22]. Our lab has already measured different molecules applying this technique [23], including 2-AG, and revealed that 3-day water-immersion stress increased 2-AG levels in the hypothalamus, midbrain, and hindbrain of senescence-accelerated mouse prone 8 mice [24].

In this study, we used DESI-MSI to investigate and reveal the 2-AG distribution in wild-type mice brain tissues and the effect of CRS on 2-AG levels in the ACC, CP, NAc, and PIR. Doing so will allow for a clearer understanding of 2-AG expression and changes in the brain and extend the horizon for future etiological and therapeutic studies of related diseases.

## 2. Materials and Methods

### 2.1. Animals

Experiments were performed on male C57BL/6JJmsSlc mice (25–30 g, 4–6 months old). They were bred in a climate-controlled standard animal house at 22 ± 2 °C with a light/dark cycle of 12 h (light period: 7 a.m. to 7 p.m.). Mice were housed two to three per cage in standard microisolator polycarbonate cages with *ad libitum* access to food and water. Subjects were randomly assigned to experimental groups, and the testing sequence was randomized throughout the experiment. All procedures were conducted according to the guidelines of the Institutional Animal Care and Use Committees of Hamamatsu University School of Medicine (HUSM), and the ethical permission number is 2022021. All efforts were made to minimize the number of animals used and their suffering.

### 2.2. Stress Protocol

Mice were subjected to stress using a published procedure [10]. In brief, mice were marked for identification on the ear as needed and randomly grouped into stress and control groups. Mice in the stress group were separately restrained for 30 min in a 50 mL-centrifuge tube with several small holes (for ventilation) in the morning for 8 consecutive days. They were placed on the bench top during the restraint period. Then we put them back in their home cage after finishing the stressor. Mice in the control group were left undisturbed in their home cages. To ensure the reproducibility of the results while complying with the requirements of the three Rs ethical principle, we used three mice in each group. All mice were weighed before inducing restraint stress on the first and last day.

### 2.3. Tissue Preparation

Mice in the stress group were immediately sacrificed by cervical dislocation after the last restraint exposure. Control animals were sacrificed immediately after removal from their home cages at the same time of day and in the same room as the stressed mice. All mice brain tissues were quickly removed and rapidly frozen in powdered dry ice, then stored at −80 °C until cryo-sectioning. The brain tissues were mounted on a sample holder with optimal cutting temperature compound (Sakura Finetek Japan, Tokyo, Japan) and coronally sectioned to a thickness of 10 µm using a cryostat system (CM1950; Leica Biosystems, Wetzlar, Germany) at −20 °C. The sliced samples were mounted on the glass slide (Matsunami, Osaka, Japan, thickness: 0.8 to 1 mm, size: 76/26 mm) for DESI-MSI and immunohistochemistry (IHC) analysis. Hematoxylin and eosin (H&E) staining was conducted after DESI-MSI measurement to facilitate anatomical localization.

### 2.4. DESI-MSI

Data were acquired within 3 days after mice were sacrificed by a Xevo G2-XS quadrupole time-of-flight (Q-TOF) mass spectrometer (Waters, Milford, MA, USA) equipped with a 2D DESI source in positive ionization mode. Prior to acquiring data, in order to obtain better signal intensity, the detector setup was performed by leucine enkephalin solution (500 µM), and the mass spectra calibration was performed by sodium formate solution (500 µM) in 2-propanol: water (90: 10, *v*/*v*). Compounds with *m/z* 443 ions from the ink of a red sharpie pen (Sharpie, USA) were used to adjust the signal intensity to over 10^6^ counts (cts) and *m/z* 798 from brain tissues over 10^4^ cts, respectively. The final optimized parameters of DESI source were as follows. The capillary voltage was set to 4.0 kV and the sampling cone voltage was set to 50 V. The source temperature was set to 130 °C and nebulizing nitrogen gas was set at a pressure of 0.5 MPa. The spray solvent (methanol: water, 98: 2, *v*/*v*) was sprayed continuously at a flow rate of 3 µL/min. The spray impact angle was 75°; the emitter tip to tissue, emitter tip to ion transfer capillary orifice, and ion transfer capillary orifice to tissue distance were approximately 2, 6, and 0.5 mm, respectively. The samples were scanned at a velocity of 200 µm/sec with a pixel size of 100 µm × 100 µm. The data were acquired in the *m/z* 100 to 1000 range; the speed of acquisition was 1 spectrum sec^−1^. The analyzer mode was set as “sensitivity” and the data type was set as “continuum”. For data processing using HDImaging software (Waters, Milford, MA, USA; version 1.4), the mass spectrum of the 1000 highest intensity peaks was collected in a mass range of *m/z* 100 to 1000, with the mass resolution set at 20,000 and a mass window of 0.02 Da.

Ions of *m/z* 309.2036 were used for lock mass correction, with a tolerance and minimum signal intensity of 0.05 amu and 500 cts, respectively. Standard 2-AG (0.1 μL) (Sigma-Aldrich, St. Louis, MO, USA) dissolved in acetonitrile at different concentrations (0 µg/mL, 1 µg/mL, 5 µg/mL, 10 µg/mL) was used to confirm the mass accuracy.

### 2.5. IHC Staining

IHC staining was conducted according to a previously described method [25], with some modifications. In brief, we proceeded step by step as follows: (1) The frozen brain sections were removed and allowed to return to room temperature (RT) and dry. (2) Then, they were fixed for 10 min in 4% paraformaldehyde and (3) blocked in 10% horse normal serum (HNS), 0.1% sodium azide, and 0.5% Triton X-100 in Tris-HCl-buffered saline (TBS) (pH 7.4) for 30 min at RT. (4) Primary antibody incubation was conducted with rabbit anti-MGL (1:500, Frontier Institute catalog #MGL-Rb, RRID: AB_2571798) on a shaker for 2 days at 4 °C. (5) Endogenous peroxidase was blocked with 3% (*v*/*v*) H_2_O_2_ for 20 min at RT. (6) The samples were incubated with goat anti-rabbit IgG (1:200, Vector Laboratories catalog #BA-1000, RRID: AB_2313606) for 1 h on a shaker at RT. (7) They were then incubated in the avidin-biotin complex (ABC, 1:50, Vector Laboratories catalog #PK-6100, RRID: AB_2336819) and prepared in a washing solution (1:50) for 30 min at RT. Between all steps, sections were washed in 1% HNS and 0.5% Triton X-100 in TBS. Visualization was performed using 0.05% diaminobenzidine (DAB) in a 0.1 M phosphate buffer solution (pH 7.6) with 0.5% Triton X-100 and 0.01% hydrogen peroxide for 3 min 30 s at RT. Finally, tissues were subsequently counterstained with hematoxylin, dehydrated in graded alcohols (80, 90, 100%), transparentized with xylene, and coverslipped with PathoMount (FUJIFILM). Controls were set up to check the accuracy.

### 2.6. Data Analysis

The general information of control and CRS mice was analyzed by the Mann-Whitney U test. Quantification of the immunoreactivity of representative areas was determined using the analysis software ImageJ (National Institutes of Health, Bethesda, MD, USA). MassLynx (Waters, Milford, MA, USA; version 4.1) and HDImaging software were used to acquire DESI-MSI data. In HDImaging software, raw data were normalized using total ion current, and ion images were constructed to visualize the spatial distributions of the detected molecules after setting parameters in visualization, including scale: linear, image smoothing: linear interpolation. The normalized spectra were exported to the MassLynx software to further normalize to the largest peak on display. MSI raw data were converted into .imzML by HDImaging software and then further converted to .imdx using an IMDX converter (Shimadzu, version 1.20.0.10960). Finally, the .imdx data were used to analyze the intensity of target *m/z* by IMAGE REVEAL software (Shimadzu, Kyoto, Japan; version 1.20.0.10960). A thresholding value was determined by the mean + SD of the control mice when comparing the 2-AG and its precursor levels between the control and stressed mice. The lower and higher intensities than the threshold value were counted and set for a 2 × 2 contingency table. Pearson’s Chi-square test was performed to analyze the MS intensity of molecules. Two-tail *t*-test was performed to evaluate the differences in fold changes in the average intensity of 2-AG and its precursors between the two groups. All statistical analyses were performed using SPSS version 22 (IBM) and GraphPad Prism version.8.0.2.263 (GraphPad Software, GraphPad Software, La Jolla, CA, USA). Differences with *p*-value < 0.05 were considered significant.

## 3. Results

### 3.1. Detection of 2-AG by DESI-MSI

After optimizing DESI-MSI parameters, the standard 2-AG was detected as potassium adduct [M + K]^+^ at *m/z* 417.2409. Consistently, the endogenous 2-AG in the mice brain sections at *m/z* 417.2409 was also detected (Figure 1A). The error in the determination of the monoisotopic mass of 2-AG was 0.48 ppm (Table 1). Ion images of different concentrations of the 2-AG standard (0 µg/mL, 1 µg/mL, 5 µg/mL, 10 µg/mL) applied to mice brain tissues were acquired (Figure 1B). We confirmed that the signal intensity of 2-AG varied depending on its concentration.

### 3.2. 2-AG Levels Are Highest in the Hypothalamus (HY) and Lowest in the Hippocampal Region (HIP)

We visualized 2-AG in the coronal brain sections of non-stressed male mice via DESI-MSI. The approximate localization of the sections from the forebrain to the cerebellum (sections I to VIII) is shown in Figure 2A. To evaluate the distribution of 2-AG in the mouse brains, we annotated the H&E stained coronal sections into detailed regions based on the Allen Brain Atlas, including the anterior commissure of the olfactory limb (aco), isocortex, striatum ventral region (STRv), caudate putamen (CP), pallidum (PAL), HY, amygdala, HIP, motor-related superior colliculus (SCm), midbrain reticular nucleus (MRN), periaqueductal gray (PAG), pons, medulla, cerebellar cortex (CBX), retrohippocampal region (RHP), and forebrain bundle system (Figure 2B). Different 2-AG signal intensities were detected in different functional areas of the brain. Moderate 2-AG signal intensity was detected in the aco, PAL, MRN, SCm, pons, and medulla. The remaining regions except for HY showed weaker signal intensity than the above-mentioned egions. The 2-AG levels were highest in the HY and lowest in the HIP (Figure 2C).

### 3.3. 2-AG Levels Were Upregulated and 2-AG Precursor Levels Were Downregulated in the ACC, CP, NAc, and PIR by CRS

The physical profiles of the control and CRS mice showed no differences in age and body weight at baseline, but the body weight on the last day of CRS exposure was significantly different (*p* = 0.046) (Table 2). Compared with the control group, CRS tended to inhibit body weight growth.

OAG and SAG are likely the predominant diacylglycerol species as 2-AG precursors in neuronal cells. They have been identified in the mice brain tissues and used for a 2-AG-related study by LC/MS [26]. The OAG or SAG can be hydrolyzed by DAGLs at the sn1 position to generate 2-AG (Figure 3A). Therefore, we also checked the distribution of these two precursor molecules. [OAG + K]^+^ (*m/z* 681.48) and [SAG + K]^+^ (*m/z* 683.50) were detected in the mice brain sections (Appendix A). Errors in the determination of the monoisotopic mass of OAG and SAG were −5.72 ppm and −1.17 ppm, respectively (Table 1).

To elucidate the effect of CRS on the distribution of 2-AG and 2-AG precursors in the different brain regions, we acquired DESI-MSI data from the coronal brain sections of control and stressed mice. Histograms were used to show the signal intensity of each pixel in the defined regions. We found that 2-AG levels were significantly (*p* < 0.05) upregulated, while OAG and SAG were significantly (*p* < 0.05) downregulated by CRS in the ACC, CP, NAc, and PIR (Figure 3B,C; Appendix A).

### 3.4. The Quantification of the Immunohistochemical Expression of MGL in the ACC, CP, NAc, and PIR Was Not Affected by CRS

As CRS caused changes in 2-AG levels in the ACC, CP, NAc, and PIR, the expression of MGL was also investigated by IHC. Antibody validation (Figure 4A) was performed in the HIP, where the expression of MGL was known [25]. MGL immunoreactivity was observed in the ACC, CP, NAc, and PIR of control and stressed mice (Figure 4B). The quantification of the immunohistochemical expression of MGL in ACC (*p* = 0.130), CP (*p* = 0.360), NAc (*p* = 0.432), and PIR (*p* = 0.5178) between control and stress groups were not different (Figure 4C).

### 3.5. CRS Upregulated 2-AG Levels in the Non-LI Area but Not in the LI of ACC

The ACC consists of layers I, II–III, and V–VI in rodents and humans [27]. LI has less cell density than non-LI and receives strong dendritic tuft branches from other layers [28]. To explore the detailed changes of 2-AG and 2-AG precursors in different neural cell density areas after CRS exposure, we analyzed the changes in these molecules in LI and non-LI areas of ACC by combining DESI-MSI image results with MGL IHC-stained results. The magnification of DESI-MSI results (Figure 5) showed that 2-AG was significantly upregulated by 12% in the non-LI areas after CRS exposure (*p* = 0.033). However, the difference between control and stress in the LI was not obvious (*p* = 0.161). OAG and SAG were significantly downregulated in both non-LI and LI of the ACC after CRS exposure. OAG decreased by 35% in non-LI (*p* < 0.001) and 33% in LI (*p* = 0.003) areas. SAG decreased by 52% in non-LI (*p* = 0.003) and 39% in LI (*p* = 0.016) areas.

## 4. Discussion

Our study shows the spatial distribution of 2-AG in control male mouse brains and reveals a moderate 2-AG signal intensity in the aco, PAL, MRN, SCm, pons, and medulla. Except for HY, the remaining regions showed weaker signal intensity than the above-mentioned regions. The HY has the highest 2-AG levels, while the HIP has the lowest. 2-AG signaling would exert different effects in different brain functional areas by targeting corresponding receptors. CB1 receptors expressed in the amygdala, cingulate cortex, prefrontal cortex, ventral PAL, CP, and NAc [29] could also be detected in the anterior commissure [30] and the majority of hypothalamic nuclei [31]. 2-AG within HY regulates the effects of time of day and stress on short-term memory [32] and plays a role in maintaining food intake [33]. In PAG, 2-AG-CB1 signaling promotes stress-induced analgesia [34]. In the hippocampus, the release of 2-AG leads to the transient suppression of gamma-aminobutyric acid-mediated transmission by retrograde signaling [35]. DAGLα [36], MGL, and CB1 receptors [25] were detectable in the hippocampus. Our results show that 2-AG distributes at a low level in the HIP, which belongs to the hippocampus, suggesting that MGL may contribute to low levels of 2-AG.

This study is the first to show that 2-AG levels were upregulated in the ACC, CP, NAc, and PIR after CRS exposure. Others have reported that 2-AG augmentation reduces stress-induced anxiety-like behavior [9] and participates in habituation to repeated stress [10]. Our DESI-MSI results provide direct evidence that 2-AG in the ACC, CP, NAc, and PIR is upregulated under CRS conditions and may be involved in habituation to stress. The CRS not only upregulated 2-AG but also downregulated OAG and SAG. DAGLα immunoreactivity has been reported in the ACC, CP, NAc, and PIR [36]. We speculate that the demand for 2-AG in the ACC, CP, NAc, and PIR increases after CRS exposure; thus, OAG and SAG are hydrolyzed more by DAGLα, leading to a decrease in them and an increase in their degradation product 2-AG. Therefore, the downregulation of OAG and SAG may be a reason for the upregulation of 2-AG after CRS exposure.

The results show that CRS tended to inhibit body weight growth. An analysis of cumulative feeding following 2 h restraint stress revealed that it has a significant effect on body weight; restraint significantly decreased intake during early post-stress time points (1 h) and overall intake at the 22 h time point [37]. It also showed that increased 2-AG following the systemic administration of an MGL inhibitor decreased energy intake. Therefore, we speculate that CRS reduced food intake to inhibit body weight growth, and CRS-induced upregulated 2-AG may play a role in this process.

Moreover, our results show that MGL expression levels in the ACC, CP, NAc, and PIR were not different between the control and stressed groups. Therefore, we suggest that the CRS-induced 2-AG increase was not caused by the change in MGL expression levels. Previous studies have reported that CB1 receptors were detected in the ACC [38,39], CP, NAc, and PIR [40]. Therefore, a functional 2-AG signaling system composed of cannabinoid receptors and the complete machinery for the synthesis and degradation of 2-AG existed in the ACC, CP, NAc, and PIR. 2-AG contributes to stress response termination by inhibiting glutamate release and restraint following anxiety arousal by stimulating presynaptic CB1 receptors [41]. 2-AG can also act on astrocytic CB1 to indirectly regulate glutamate release via the release of gliotransmitters [42]. These reports and our results suggest that CRS-induced upregulated 2-AG in the ACC, CP, NAc, and PIR plays a role in contributing to stress response termination by regulating glutamate release through targeting presynaptic or astrocytic CB1 receptors.

The DESI-MSI results also show that CRS significantly upregulated 2-AG levels in the non-LI area but not in LI of the ACC. The MGL result shows that LI has fewer neural cells than non-LI areas in ACC. LI contains interneurons; non-LI contains both pyramidal neurons and interneurons [27]. 2-AG changes in the ACC caused by CRS may be more concentrated in pyramidal neurons. 2-AG might be involved in antidepressant mechanisms by targeting CB1 receptors in neurons and astrocytes and CB2 receptors in microglial cells [7]. 2-AG could also avoid the detrimental effects of inflammation through microglia [43]. Thus, we supposed that the increased 2-AG by CRS might be involved in antidepressant and anti-neuroinflammation.

In addition, it is important to note that excessive 2-AG inevitably increases its degradation by MGL to generate neuroinflammatory prostaglandins that promote neuroinflammation [44]. Therefore, a prolonged stressor with persistently elevated 2-AG degradation products is not beneficial for neurodegenerative disorders. Interestingly, the inhibition of MGL would promote the augmentation of the endogenous 2-AG that can be therapeutically useful for suppressing neuroinflammation and maintaining the balance between excitatory and inhibitory neurotransmission [45]. Therefore, our results on MGL expression and 2-AG changes also lay the foundation for the future use of MGL inhibitors to achieve antidepressant and anti-neuroinflammation effects.

Furthermore, the bilateral injection of 2-AG into the NAc shell produced a clear-cut, short-term stimulatory action on feeding behavior [46]. The increase of 2-AG in NAc due to stress may explain stress-induced overeating. 2-AG in the ACC likely modulates fear-conditioned analgesia, and 2-AG-CB2 receptor signaling may suppress this form of endogenous analgesia [47]. 2-AG also targets GPR55, and the activation of GPR55 signaling in the ACC facilitates inflammatory pain via a top-down modulation of descending pain control [48]. Thus, the CRS-induced elevated 2-AG in the ACC may also contribute to stress-induced analgesia or hyperalgesia processes through CB2 or GPR55 receptors.

To ensure the accuracy of the 2-AG measurements, we not only calibrated each measurement with standard 2-AG on each slide but also took special care to minimize post-mortem changes and degradation of 2-AG over time throughout the process. The steps and times were consistent from sample to sample, and the exposure time from −80 °C to −20 °C was consistent across samples and as minimal as possible during cryo-section. Moreover, we avoided any interference of drugs or injection stress because it has been indicated that morphine [49] and fentanyl [50] downregulate the contents of 2-AG. Hence, we chose cervical dislocation to sacrifice mice, a method already used in 2-AG-related animal experiments [51]. Neuronal localization of putative 2-AG was achieved by space and time coherent mapping MSI [52]. The three-dimensional distribution of 2-AG in the brain tissue can be achieved in the future by DESI-MSI, as with previous research on the three-dimensional distribution of monoamines in mouse brains [53]. DESI-MSI is a powerful method that has the potential to achieve further progress in 2-AG-related studies.

In summary, we suggest that the CRS-induced elevated 2-AG in ACC CP, NAc, and PIR constitutes self-regulation during habituation to stress. The elevated 2-AG in these regions may play a role in stress response termination, antidepressant and anti-neuroinflammation effects, and stress-induced analgesia or hyperalgesia. An imbalance of 2-AG in the brain seems to trigger or promote the development of stress-related neuropsychiatric diseases and neuroinflammation-related neurodegenerative disorders. The results of this study extend the horizon for the etiological investigation or treatment of these related diseases.

## 5. Conclusions

The results presented in this study show that 2-AG levels are highest in the HY and lowest in the HIP. Moreover, this study reveals that CRS upregulates 2-AG levels and downregulates its precursors’ levels in the ACC, CP, NAc, and PIR. The expression levels of MGL were not affected by CRS. Our findings extend the horizon for future etiological and therapeutic studies of 2-AG-related diseases, such as stress-related neuropsychiatric diseases.

## Figures and Tables

**Figure 1 cells-12-00393-f001:**
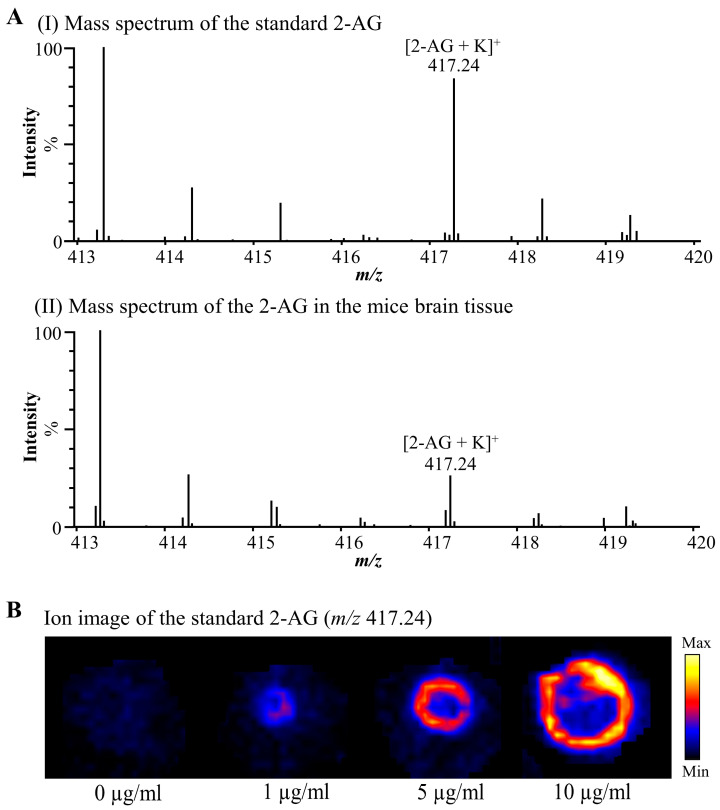
2-AG was detected from standard 2-AG and male mice brain tissue by DESI-MSI in positive ion mode. (**A**) DESI-MSI mass spectra acquired from (I) standard 2-AG applied to mice brain sections and (II) male mice brain tissues. (**B**) DESI-MSI ion image of different concentrations of standard 2-AG applied to mice brain sections. The concentrations from left to right are 0 µg/mL, 1 µg/mL, 5 µg/mL, and 10 µg/mL. 2-AG: 2-arachidonoylglycerol.

**Figure 2 cells-12-00393-f002:**
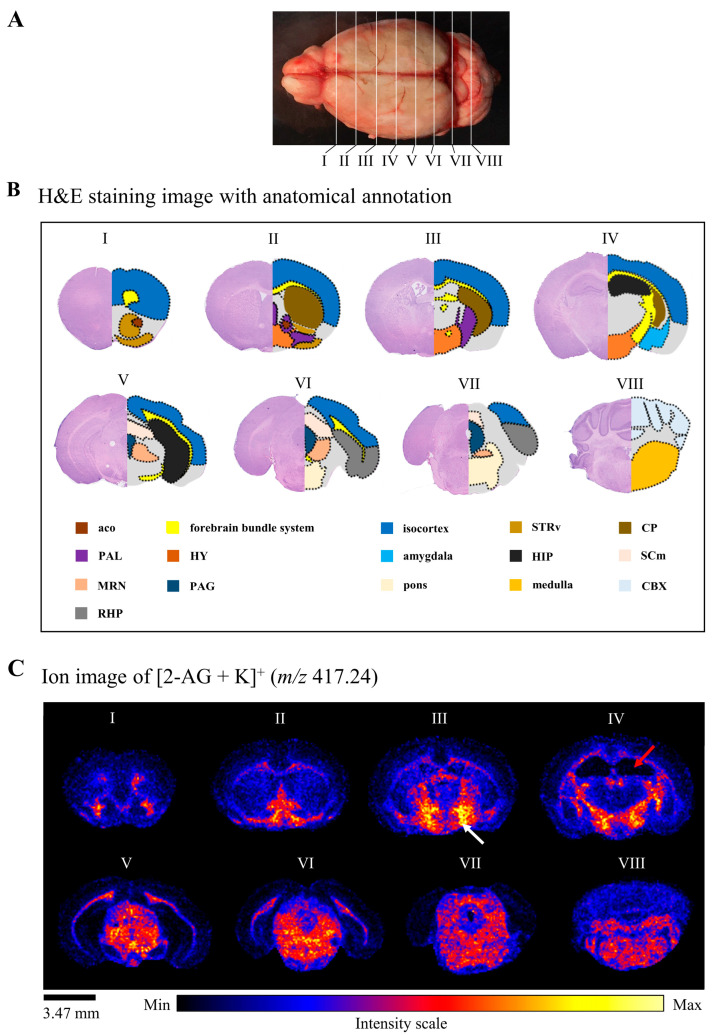
2-AG levels are highest in the HY and lowest in the HIP from the forebrain to the cerebellum. (**A**) Schematic of the location of the sections (from I to VIII) from the forebrain to the cerebellum. The interval between each section is approximately 1200 µm. (**B**) H&E images (I to VIII) with anatomical annotation, following Allen Brain Atlas (https://atlas.brain-map.org/, accessed on 1 November 2022). (**C**) Ion images of 2-AG distribution in corresponding coronal brain sections (I to VIII). The 2-AG signal intensity of the corresponding area is displayed according to the rainbow colors. The white and red arrows indicate the highest and lowest 2-AG levels, respectively (scale bar: 3.47 mm). aco: anterior commissure, olfactory limb, STRv: striatum ventral region, CP: caudate putamen, PAL: pallidum, HY: hypothalamus, HIP: hippocampal region, SCm: superior colliculus, motor-related, MRN: midbrain reticular nucleus, PAG: periaqueductal gray, CBX: cerebellar cortex, RHP: retrohippocampal region.

**Figure 3 cells-12-00393-f003:**
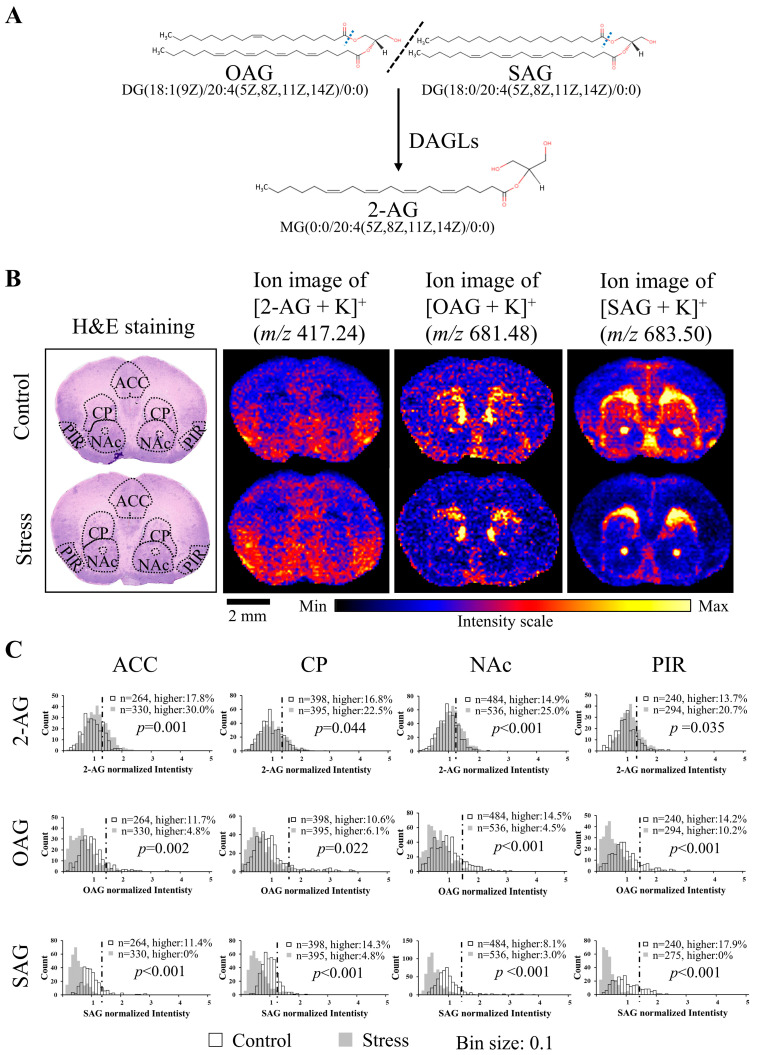
2-AG levels were upregulated and the levels of OAG and SAG were downregulated in the ACC, CP, NAc, and PIR by CRS. (**A**) Relationship between 2-AG and OAG, SAG. The common name and structure of 2-AG, OAG, and SAG are cited from the Human Metabolome Database (https://hmdb.ca/, accessed on 1 November 2022). (**B**) H&E staining results and ion images of 2-AG, OAG, and SAG in coronal brain sections of control and stressed mice (scale bar: 2 mm). ACC, CP, NAC, and PIR in control and stressed mice brains were circled by a black dotted line for analysis. Anatomical annotations followed Allen Brain Atlas. (**C**) Histograms represent the intensity change of 2-AG, OAG, and SAG in the ACC, CP, NAC, and PIR of control and stressed mice. The value of each pixel in the stress and control groups was normalized by dividing the mean of the control group. The horizontal axis of the histograms represents the normalized signal intensity level, and the vertical axis indicates the frequency of pixels. A threshold line was set as mean + SD of the control mice. Pearson’s Chi-square test was performed for lower-than-threshold and higher-than-threshold signals in these two groups (n: the number of pixels). OAG: 1-oleoyl-2-arachidonoyl-sn-glycerol, SAG: 1-stearoyl-2-arachidonoylglycerol, DAGL: diacylglycerol lipases, ACC: anterior cingulate cortex, CP: caudate putamen, NAc: nucleus accumbens, PIR: piriform cortex.

**Figure 4 cells-12-00393-f004:**
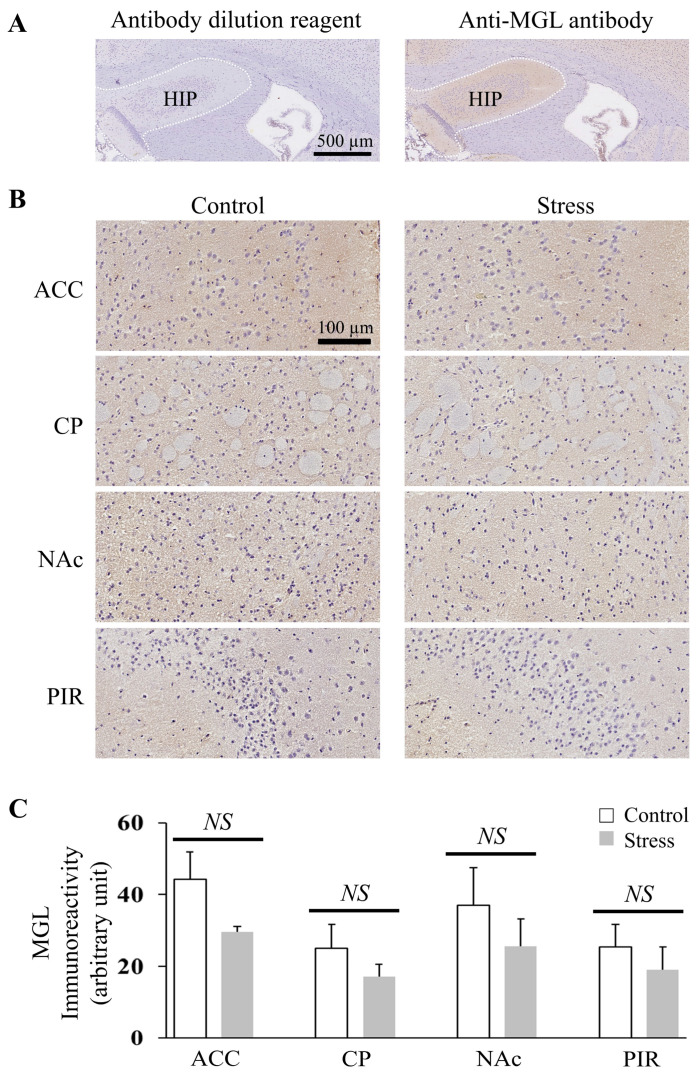
MGL immunoreactivity in the ACC, CP, Nac, and PIR was not different between control and stressed mice. (**A**) Immunohistochemical staining for the validation of MGL in mice HIP (the white dotted line circled part). The left is the result of the antibody dilution reagent and the right is the result of MGL (500X) (scale bar: 500 µm). (**B**) Representative MGL-immunoreactive result of ACC, CP, NAc, and PIR in the control and stressed mice sections (scale bar: 100 µm). (**C**) Quantification comparison of the immunohistochemical expression of MGL in the ACC, CP, NAc, and PIR between control and stressed mice. Histograms represent the mean + SEM (the number of mice in each group was three). Two-tail t-test was performed. NS: *p*-value > 0.05.

**Figure 5 cells-12-00393-f005:**
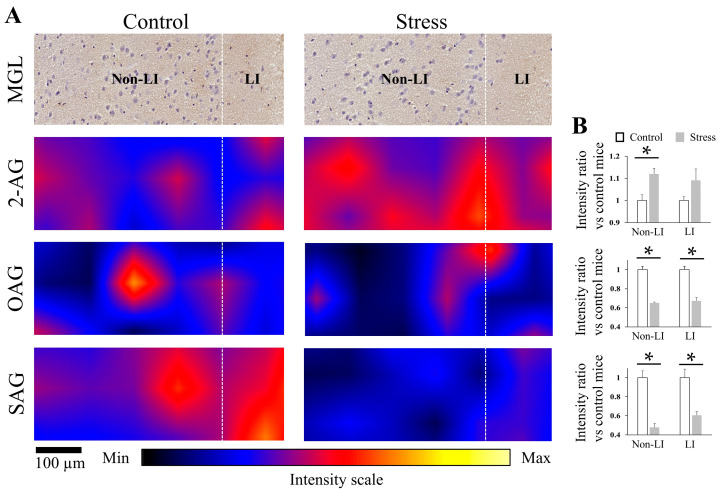
Magnification of 2-AG, OAG, and SAG expression in the LI and non-LI of ACC in control and stressed mice. (**A**) Representative images of ACC section immunostained for MGL and corresponding DESI-MSI results of 2-AG, OAG, and SAG in control and stressed mice (scale bar: 100 µm). (**B**) Fold changes in the average intensity of 2-AG, OAG, and SAG in the LI and non-LI of the ACC sections of stressed mice compared to those of control mice (the number of mice in each group was three). Two-tail *t*-test was performed. *: *p* < 0.05. LI: layer I, non-LI: the layers of ACC without layer I.

**Table 1 cells-12-00393-t001:** *m/z* accuracy of targeted molecules from standard 2-AG and mice brain tissues.

Assigned Molecule	Theoretical *m/z*	Observed *m/z*	Mass Accuracy (ppm)
[2-AG + K]^+^	417.2407	417.2409	0.48
[OAG + K]^+^	681.4860	681.4821	−5.72
[SAG + K]^+^	683.5017	683.5009	−1.17

**Table 2 cells-12-00393-t002:** Summary of the general information of control and stressed mice.

Group	Control	Stress	*p*-Value
Age (day)	165 ± 13	160 ± 13	NS
BW (g) of day 1	27.03 ± 0.28	26.40 ± 0.21	NS
BW (g) of day 8	28.03 ± 0.43	25.73 ± 0.32	0.046

The number of mice in each group was three, NS: *p*-value > 0.05, BW of day 1: body weight on the first day of stress exposure, BW of day 8: body weight on the last day of stress exposure. Values are mean ± SEM. Mann-Whitney U test was performed.

## Data Availability

All relevant data were reported within the article. Further supporting data will be provided upon a written request addressed to the corresponding author.

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
