# Peer review of "Endocannabinoid 2-Arachidonoylglycerol Levels in the Anterior Cingulate Cortex, Caudate Putamen, Nucleus Accumbens, and Piriform Cortex Were Upregulated by Chronic Restraint Stress"

_cells, 2023, doi:10.3390/cells12030393_

Round 1

Reviewer 1 Report

Overall, a well prepared and performed study.  The MS imaging is of high quality.

Why N=3?  This appears a small number based on the variability  expected, and certainly compared to most prior studies on stress and brain neurochemistry.

Given the use of three animals for the two conditions, the  other statistical tests appeared strange.  As an example, the Pvalues in the DESI images (e.g. Figure 3 and SI images) are a little misleading – they were done on a per pixel basis, and of course, there are hundreds of pixels so the Pvalues look impressive.  However, from a biological context (which is what the manuscript is about),  shouldn't discussion on regional changes be done between the biological replicates (which is an n of 3) and not using the individual pixels as the n’s for the pvalue calculations?  While this sued to be common in MSI, the use of each tissue images as a single replicate is a much better practice and brings MSI into line with other imaging approaches.   

The manuscript has the following statement: "Our lab has already measured different molecules applying this technique [23], in-80 including 2-AG [24]."  When looking at this citation, it is actually more than visualizing the same molecules, but involves a study on how stress upregulates 2-AG in several brain regions… as imaged by DESI-MS.  In other words, the prior recent article is surprisingly close including the ssame molecules, stress, mouse brains, etc. It is true that the current article includes greater detail on localization and a simpler stress protocol.  The authors need to discuss the prior work and put it into context of the new work and highlight what the differences are between the two. 

Author Response

Thanks again for your comments and suggestions.

Reviewer 2 Report

1>In your researcher, Have you considered the gender factor will affect 2-AG spatial distribution?

2>In line 225-230, the result shows the body weight growth by CRS. Is that caused by reduced food intake? Please give more detailed analysis.

3>Please indicate  when you get the 2-AG images (how long after CRS?) to make the results more convinced.

Author Response

(The authors gave the same response as above.)

Reviewer 3 Report

This is a straightforward and well-written manuscript.  The authors address the issue of the effect of chronic restraint stress on cannabinoid levels in the brain. The focus is on 2-AG because it has been shown previously to alleviate stress-induced anxiety-like behavior.  Here, the authors show that chronic restraint stress increases 2-AG levels in several brain regions. The increase is prominent in the hypothalamus and almost absent in the hippocampus.  With an increased demand for 2-AG, its precursors OAG and SAG are found at lower levels. This downregulation of OAG and SAG might be the reason for the upregulation of 2-AG after chronic restraint stress.

Figure legend 2: I see a white and red arrow but not a black arrow.

The authors detected low levels of 2-AG in the hippocampus.  Is it possible that cells in the hippocampus use anandamide instead of 2-AG?

Author Response

(The authors gave the same response as above.)
